# Application of the Nutrient-Rich Food Index 9.3 and the Dietary Inflammatory Index for Assessing Maternal Dietary Quality in Japan: A Single-Center Birth Cohort Study

**DOI:** 10.3390/nu13082854

**Published:** 2021-08-19

**Authors:** Chihiro Imai, Hidemi Takimoto, Ayako Fudono, Iori Tarui, Tomoko Aoyama, Satoshi Yago, Motoko Okamitsu, Satoshi Sasaki, Shuki Mizutani, Naoyuki Miyasaka, Noriko Sato

**Affiliations:** 1Department of Molecular Epidemiology, Medical Research Institute, Tokyo Medical and Dental University, Tokyo 113-8510, Japan; imai.epi@mri.tmd.ac.jp; 2Department of Nutritional Epidemiology and Shokuiku, National Institutes of Biomedical Innovation, Health and Nutrition, Tokyo 162-8636, Japan; thidemi@nibiohn.go.jp (H.T.); cynthiasnow90@gmail.com (I.T.); tomokom@nibiohn.go.jp (T.A.); 3Comprehensive Reproductive Medicine, Graduate School of Medical and Dental Sciences, Tokyo Medical and Dental University, Tokyo 113-8510, Japan; faya.per@tmd.ac.jp (A.F.); n.miyasaka.gyne@tmd.ac.jp (N.M.); 4Child and Family Nursing, Graduate School of Health Care Sciences, Tokyo Medical and Dental University, Tokyo 113-8510, Japan; sycfn@tmd.ac.jp (S.Y.); motoko.cfn@tmd.ac.jp (M.O.); 5Department of Social and Preventive Epidemiology, School of Public Health, The University of Tokyo, Tokyo 113-0033, Japan; stssasak@m.u-tokyo.ac.jp; 6Institute of Advanced Biomedical Engineering and Science, The Public Health Research Foundation, Tokyo 169-0051, Japan; skkmiz@gmail.com

**Keywords:** maternal dietary quality, NRF9.3, DII, pregnancy, DOHaD

## Abstract

The maternal diet can potentially influence the life-course health of the child. A poor-quality maternal diet creates nutrient deficiencies and affects immune–metabolic regulation during pregnancy. The nutrient-based overall dietary quality can be assessed using the Nutrient-Rich Food Index 9.3 (NRF9.3), which measures adherence to the national reference daily values of nutrient intake. Pro- and anti-inflammatory nutrient intake can be assessed using the energy-adjusted dietary inflammatory index (E-DII), a comprehensive index of diet-derived inflammatory capacity. Using these indices, we assessed the overall dietary quality and inflammatory potential of pregnant women during mid-gestation in an urban area of Japan (*n* = 108) and found that there was a strong inverse correlation between the NRF9.3 and E-DII scores. Comparison of the scores among the tertiles of NRF9.3 or E-DII indicated that dietary fiber, vitamin C, vitamin A, and magnesium mainly contributed to the variability of both indices. Intake of vegetables and fruits was positively associated with high NRF9.3 scores and negatively associated with high E-DII scores, after adjustment for maternal age, pre-pregnancy body mass index, and educational level. Consistent with the previous studies that used dietary pattern analysis, this study also demonstrated that vegetables and fruits were the food groups chiefly associated with high dietary quality and low inflammatory potential among pregnant Japanese women.

## 1. Introduction

Multiple lines of evidence from epidemiological observations have implicated that the early-life environment is linked to the risk of noncommunicable diseases later in life [1,2,3]. The Developmental Origin of Health and Disease (DOHaD) determined that the developing conditions in utero modify the long-lasting bodily functions and physiology of the offspring [3]; thus, the maternal diet can influence the life-course health of the child.

Insufficient maternal dietary intake that does not meet the increased demands during pregnancy is a risk factor for adverse birth outcomes, such as low birth weight, preterm birth, and intrauterine growth restriction [4]. Previous studies have focused on the relationship between the intake of specific nutrients and newborn height and weight [5,6]. Appropriate maternal intake of methyl-donor nutrients, micronutrients, and omega-3 fatty acids is particularly important for fetal neurodevelopment [7,8].

Recently, comprehensive assessments of maternal diets have been conducted using the Healthy Eating Index, Alternate Healthy Eating Index, and Dietary Approaches to Stop Hypertension (DASH) [9,10,11], as well as by the application of dietary pattern analysis to identify the effects of diet on various pregnancy outcomes [12,13,14,15,16]. However, these studies have mainly focused on certain foods or food groups. Moreover, studies involving comprehensive dietary quality assessments based on multiple nutrients, rather than an individual nutrient, are less common.

Both food-based and nutrient-based dietary indices are useful in assessing the overall quality and/or properties of a pregnant woman’s diet. However, the international use of food-based indices will require harmonization of the food database because foods frequently consumed are unique to each country’s dietary culture. In contrast, nutrients are universal so that nutrient-based dietary indices are globally used without further processing. Therefore, in this study, we used two dietary indices that are commonly used in the world, which cover a wide range of nutrients: the Nutrient-Rich Food Index 9.3 (NRF9.3) and the energy-adjusted dietary inflammatory index (E-DII) (Figure 1).

The NRF9.3 was originally developed to score the nutritional value of foods [21] and has recently been used to assess the nutritional value of the diet consumed by individuals [22]. The NRF9.3 is useful for assessing the overall dietary quality; however, it has not yet been applied to assess the dietary quality of pregnant women.

The DII is a comprehensive index of diet-derived inflammatory capacity [19] and was designed to be universally applicable across all human studies for dietary assessment [20,23]. Since the relationship between energy and nutrient consumption varies across populations, the energy-adjusted DII (E-DII) was developed, which is often used as an improved version of DII [20]. Multiple studies have reported that E-DII scores are not only associated with cardiovascular disease, obesity [24], and inflammatory biomarkers [25,26] but also maternal and child health [11,27,28]. It has also been documented that the unadjusted DII score is linked to the risk of preterm birth and low birth weight [29]. However, with the exception of one study [30], all previous investigations have focused on the total E-DII (or DII) score and do not identify which parameters contribute most to the variability of DII scores in each analyzed population.

Both NRF9.3 and E-DII are nutrient-based dietary metrics. We applied them to pregnant women, because the main nutritional problems during pregnancy are nutrient intake below the requirements for pregnancy and/or excessive intake of nutrients associated with low-grade chronic inflammation [31].

In this study, we aimed to assess and report the overall dietary quality and inflammatory potential of pregnant women in a single-center birth cohort in Japan using the NRF9.3 and the E-DII. We also intended to the nutrients that contribute to the variability of each score and the relationship between these scores and the intake patterns of food groups.

## 2. Materials and Methods

### 2.1. Study Population and Design

The current research was based on a prospective mother-child cohort study in the Metropolitan Tokyo area: the Birth Cohort Gene and ENvironment Interaction Study of Tokyo Medical and Dental University (TMDU) (BC-GENIST), which was designed to evaluate the effects of the prenatal environment and genotype on the epigenetic state of mothers and their offspring [32]. During the years 2015–2019, pregnant women in their first trimester were recruited to BC-GENIST at the TMDU hospital (*n* = 126). All the BC-GENIST participants provided their written informed consent. The participants who later withdrew their consent (*n* = 4) and those who did not give birth at the TMDU hospital (*n* = 7) were excluded. Those with multiple pregnancies (*n* = 3), non-Japanese subjects (*n* = 2), participants with insufficient food records (*n* = 1), and participants who were hospitalized until delivery (*n* = 1) were also excluded. A total of 108 participants were eligible for this study. The study was approved by the Ethics Review Committee of the Faculty of Medicine and the Medical Research Institute of TMDU (G2000-181).

### 2.2. Data Collection

In this cohort study, non-consecutive three-day dietary records were obtained from the study participants during the mid-gestation period (the median value [interquartile range] of the period for data collection was 19 weeks; range, 17–22 weeks). Upon collection of the dietary records, they were checked by the interviewers to clarify any ambiguous points. Because the participants were recruited sequentially during prenatal checkups, the months in which the records were collected varied (22% in March to May; 30% in June to August; 12% in September to November; 36% in December to February). Information on smoking (smoking during pregnancy or not), educational level (university degree or higher/lower), and economic status (household income above/below 6,000,000 JPY) was collected via a self-administered lifestyle questionnaire. Data on the maternal age, height, pre-pregnancy weight, parity, and fetal sex were collected from the medical records.

### 2.3. Dietary Data

Dietary intake was assessed from the three-day food dietary records. The participants were provided with a notebook to record the amount and type of food and beverages consumed. Trained interviewers/dietitians instructed the participants on how to complete the dietary records for three non-consecutive days. The participants were advised to avoid recording intake of weekends and holidays. Approximately four weeks later, the participants returned the notebook to be checked by the study dietician. The dieticians used a photobook of 122 commonly eaten foods and dishes [33] to identify the portion sizes of the foods and beverages consumed. The pictures in this photobook were real-sized, accompanied by weight (g) of the food. 

Upon collection of the food dietary records, all the recorded sheets were checked by dietitians to clarify any ambiguous points, such as adding seasonings at the table, consumption of tea or other beverages, and snacking between meals. The dieticians then converted the portions of the foods consumed into estimated intake (g). When a raw food such as vegetables, meat, fish, or egg was heat-cooked, the type of cooking was also checked and recorded. Each food item was coded according to the food numbers in the Standard Tables of Food Composition (STFC) in Japan, 2015 [34] so that energy and nutrient intake could be calculated. Food group intake was calculated according to the food group classification in the National Health and Nutrition Survey (NHNS) [35]. Added sugar intake was calculated based on a recently developed comprehensive composition database by subtracting the total sugar contents derived from fruit juices from the free sugar contents [36]. Sugar-sweetened beverages in this study included soft drinks, sports drinks, fruit drinks, milk-based beverages, cocoa, sugar-sweetened tea, and sugar-sweetened coffee. The individual’s usual intake of nutrients and foods was estimated using the mean values over three days. In this study, dietary intake from supplements was not considered as our intention was the assessment of dietary quality from foods and beverages.

### 2.4. NRF9.3

Overall dietary quality was assessed using the NRF9.3, as described in previous studies [21,37]. In brief, the NRF9.3 score was calculated as the sum of the percentage of reference daily values (RDV) for nine qualifying nutrients (protein, dietary fiber, vitamin A, vitamin C, vitamin D, calcium, iron, potassium, and magnesium) minus the sum of the percentage excess of RDV for three disqualifying nutrients (added sugars, saturated fatty acids, and sodium). The NRF9.3 has been employed in various countries to assess the overall dietary quality by using the national dietary standards for sex and age as the RDV [17,18]. 

Age-specific RDVs for pregnant women at mid-gestation were determined based on the Dietary Reference Intakes (DRIs) for the Japanese population, 2015 [38] as shown in Appendix A. The 2015 version of the DRI was applied, because the participant recruitment took place from 2015 to 2019. For six nutrients (protein, vitamins A and C, calcium, iron, and magnesium), the recommended dietary allowance (RDA) was indicated in the DRIs, the RDA was used as the RDV for those nutrients. For vitamin D, adequate intake (AI) was used as RDV, because RDA for vitamin D was not provided in DRIs. For dietary fiber, potassium, saturated fatty acids, and sodium, the tentative dietary goal for preventing lifestyle-related diseases (DG) is indicated in the DRI, therefore, DG was used as RDV for these nutrients. For added sugars, the conditional recommendation advocated by the World Health Organization (i.e., upper limit of 5% of energy) [39] was used owing to the lack of a recommended value for added sugar in Japan as well as its low intake [36,40].

The NRF9.3 component and total scores were calculated based on the overall daily intake of each nutrient for each participant, which was adjusted for energy intake (EI) by the density method and then normalized for age-specific Estimated Energy Requirement (EER) for pregnant women at mid-gestation with a moderate level of physical activity [38]. In the NR calculation, the normalized daily intake was expressed as a ratio of RDV for each qualifying nutrient. The NR ratios were truncated at 1, so that an excessively high intake of one nutrient could not compensate for the dietary inadequacy of another. In the limiting nutrient (LIM) calculation, excessive levels of RDVs were considered. The sum of the NR of the nine qualifying nutrients minus the sum of the LIM of the three disqualifying nutrients was multiplied by 100 to yield the NRF9.3 score. Higher NRF9.3 scores indicate better nutrient adequacy, and a maximum possible score of 900 suggests a diet in which intake per given amount of energy is above the RDVs for the nine qualifying nutrients and are below the RDVs for the three disqualifying nutrients.

### 2.5. E-DII (Energy Adjusted-DII)

Maternal dietary property was assessed using the E-DII, which characterizes the dietary inflammatory potential [19,20]. The DII algorithm is the weighted sum (weights are literature-derived inflammatory effect scores) of the standardized values for each individual’s intake of a specific parameter using a common global mean and standard deviation (SD) deduced from the globally representative world database. DII parameters include 26 nutrients, 18 foods, and energy intake [19]; however, the number of DII parameters available varied depending on the region and the dietary research methods. Therefore, many studies have relied upon only 20–30 components, mainly nutrients, to calculate the DII index [11,28], because the association between the DII and inflammatory biomarkers is not affected by a reduction in the number of components [41]. From the original 45 parameters, we excluded 18 parameters, including alcohol, caffeine, and other parameters which were not available in the STFC in Japan [34]. Accordingly, 27 food parameters, including 7 pro-inflammatory nutrients (protein, total fat, saturated fatty acids, cholesterol, carbohydrates, iron, vitamin B12) and 20 anti-inflammatory nutrients and foods (monounsaturated fatty acids [MUFAs], polyunsaturated fatty acids [PUFAs], n-3 fatty acids, n-6 fatty acids, magnesium, zinc, selenium, vitamin A, β-carotene, vitamin D, vitamin E, vitamin B1, vitamin B2, niacin, vitamin B6, folate, vitamin C, dietary fiber, garlic, and onion) were used to calculate the DII scores in this study (Appendix A). 

Among the nine NRF9.3 qualifying nutrients, five nutrients (dietary fiber, vitamin A, vitamin C, vitamin D, and magnesium) are included as anti-inflammatory nutrients in the DII index. To control for the effect of total EI, the E-DII was calculated. All nutrient data in the dietary records and the global dietary intake database were converted to values per 1000 kcal by dividing these data by the EI from the diet and multiplied by 1000. This energy-adjusted individual dietary intake was standardized as a Z-score using the standard global mean from the reported amount and dividing this value by the global SD. The Z-score of each value was converted to a percentile score, which was then doubled, and then 1 was subtracted. Next, the centered value was multiplied by the respective overall food parameter-specific inflammatory effect score. These parameter-specific E-DII scores were summed to create the overall E-DII score for each participant. Higher E-DII scores represent more pro-inflammatory dietary potential, while lower E-DII scores represent more anti-inflammatory dietary potential. The theoretical full range of DII score using the 45 parameters is −8.87 to 7.98, while the score using the 25–30 parameters usually falls in the range of −5.5 to 5.5 [20]. The association between the E-DII score and inflammatory markers has been previously validated in the Japanese population [25].

### 2.6. Nutrient Intake Comparison between the BC-GENIST Participants and the NHNS Pregnant Women Cohort

The pregnant women cohort data of the National Health and Nutrition Survey (NHNS) results (2015–2017) were obtained from https://www.nibiohn.go.jp/eiken/kenkounippon21/en/eiyouchousa/ (accessed on 13 May 2021), to compare the overall nutritional characteristics between the BC-GENIST participants and the NHNS pregnant women cohort.

### 2.7. Statistical Analysis

R software (version 4.0.3) and SPSS statistical software (version 24; IBM Corp., Armonk, NY, USA) were used for the statistical analyses. To minimize the influence of dietary misreporting, the nutrients and the food group intake variables were adjusted with total energy by the density method [42]. For the correlation analysis between the nutritional scores, Spearman’s correlation coefficient was calculated. Differences in means across various classification groups were tested using one-way analysis of variance (ANOVA) for continuous variables and Chi-square test for categorical variables. The trends (increase or decrease) in food group intake according to the tertile category of dietary index were examined using general linear models with adjustment for variables previously reported to affect food intake of pregnant women (maternal age [continuous], pre-pregnancy body mass index [continuous], and educational attainment [dichotomous categorical variable]) [12,43]. Household income was a potential confounder, but in our cohort, household income was not associated with dietary indices or food intake analyzed. A previous study on a Japanese population also reported that household income was not associated with any nutrients or foods examined [44]. Therefore, household income was not included as an adjustment factor in this analysis. All reported *p*-values were two-tailed, and *p*-values < 0.05 were considered statistically significant.

### 2.8. Misreported Energy Intake (EI) and Sensitivity Analysis

To ascertain the adequacy of the EI reporting, the ratio of the reported EI to the EER was calculated for the individual participants. The individual EER was calculated as the sum of the product of basal metabolic rate (BMR) and the physical activity level (PAL) and the additional requirement specific to pregnant women at mid-gestation. BMR was computed using the Ganpule equation [45] and a PAL of 1.6 (moderate physical activity level) [38] was employed for the calculations for pregnant women at mid-gestation. The additional requirement at mid-gestation was 250 kcal/day [38]. To identify physiologically implausible self-reported EIs, 95% confidence limits of agreement were determined for the ratio of reported EI to EER using the Goldberg method [46,47]. The 95% confidence limits ±2 SD cut-offs were derived using the following equation:$$95\%\;Confidence\;limits=\pm 2\times \sqrt{\frac{CV_{EI}^{2}}{d}+CV_{TEE}^{2}+CV_{pER}^{2}}$$
where, *CV_EI_* is the within-person coefficient of variation in the reported EI, *d* is the number of days of dietary assessment, *CV_TEE_* is the day-to-day variation in total energy expenditure, and *CV_pER_* is the error in predicated energy requirements.

In our cohort, the *CV_EI_* was 17%. For *CV_TEE_* and *CV_pER_*, we used the values of 9.6% and 10.9%, respectively, which were applied to pregnant women by Nowicki et al. [47]. The deduced 95% confidence limits were found to be ±35%. Therefore, the participants with reported EI:EER ratios of 0.65–1.35 were considered acceptable reporters and those with EI to EER ratios of <0.65 were considered under-reporters in this study.

The mean (SD) of the EI:EER ratio, a measure of EI misreporting, was 0.81 (0.15). Among the 108 participants, 17 women were under-reporters, and 91 women were acceptable reporters of EI.

Sensitivity analysis was performed excluding 17 participants with misreported energy intake.

## 3. Results

### 3.1. Characteristics of Participants According to Tertile Category Groups of Each Dietary Index

The mean (SD) NRF9.3 score was 602 (106), and the range was from 333 to 807. The mean (SD) E-DII score was 1.00 (1.55), which ranged from −3.36 to 4.23. The distributions of the NRF9.3 and E-DII scores are shown in Figure 2. According to each dietary index score, the participants were stratified by tertiles. The NRF9.3 scores across the tertiles were 333 ≤ T1 ≤ 565; 570 ≤ T2 ≤ 655; 655 < T3 ≤ 807. The E-DII scores across the tertiles were −3.36 ≤ T1 ≤ 0.33; 0.39 ≤ T2 ≤ 1.86; 1.93 ≤ T3 ≤ 4.23. There was a strong inverse correlation between the NRF9.3 total scores and the E-DII total scores (*r* = −0.793, *p* < 2.2 × 10^−6^).

In Table 1, the characteristics of the BC-GENIST participants are presented according to the tertile classification of each dietary index. Among the tertiles of both indices, there were no significant differences in any of the maternal characteristics investigated. No participants smoked during pregnancy.

We compared the nutrient intake of our cohort with the results of the NHNS pregnant women cohort [35] (Appendix A). The nutrient intake levels of the participants in our cohort were similar to those in the national survey, except for the fact that the saturated fatty acids intake was higher (~20% higher than the mean of NHNS data) in our cohort.

### 3.2. Breakdown of NRF9.3 Score into Component Scores by Tertile Groups

The NRF9.3 score is the NR sub-score of the sum of the nine qualifying nutrient scores minus the LIM sub-score of the sum of the three disqualifying nutrients. We compared each component score among the tertile groups to identify the nutrients contributing to the variability in the NRF9.3 score (Figure 3). In the NR sub-scores, the protein component scores were high in all the tertile groups, but the other components increased in the order of T1, T2, and T3. For the LIM sub-scores, the sodium scores did not differ across tertiles. The scores for added sugars and saturated fatty acids tended to be lower in T3, but large variations (large SD) were observed in the scores for all groups. The main nutrients contributing to the increase or decrease in the total NRF9.3 scores were dietary fiber, iron, potassium, magnesium, and vitamin C.

We further compared the proportion (%) of participants whose daily intake was lower than the Estimated Average Requirement (EAR) or the AI across the tertiles of NRF9.3 scores (Appendix A). Iron intake was lower than the EAR for 100% of the subjects in all groups. A small number of subjects had a selenium intake lower than the EAR in all groups. For the other nutrients investigated, the proportion decreased as the NRF9.3 score increased in the order of T1, T2, T3. In the poorest dietary group, the T1 group, over 80% of the participants had an intake of vitamins A, D, E, B1, B2, B6, C, folate, calcium, magnesium, and iron that was below the EAR (AI).

### 3.3. Breakdown of E-DII into Parameter-Specific Scores by Tertile Groups

The E-DII score is the sum of scores specific to anti-inflammatory and pro-inflammatory parameters. The higher E-DII scores indicate higher inflammatory potential, which is not a desirable dietary state. In Figure 4, parameter-specific E-DII scores are shown. For example, the intake of dietary fiber, one of the anti-inflammatory nutrients, was much lower than the global mean, which resulted in a large positive parameter-specific score. We compared each parameter-specific score among the tertile groups to identify the parameters that contribute to the variability in E-DII. In contrast to the NRF9.3 score, a higher E-DII score indicates a lower dietary quality with more inflammatory potential. The mean scores of pro-inflammatory nutrients were low and not different among the tertile groups of E-DII scores. In contrast, the mean scores of most anti-inflammatory nutrients were high and increased in the order of T1, T2, T3. The main nutrients contributing to the increase or decrease in total E-DII scores were dietary fiber, vitamin A, niacin, vitamin E, β-carotene, magnesium, vitamin B1, vitamin C, zinc, vitamin B6, and folate.

The proportion (%) of the participants whose daily intake was less than the EAR or the AI across tertiles of E-DII was also investigated (Appendix A). Iron intake was found to be lower than the EAR for 100% of the participants in all groups. A small number of participants in each group had a selenium intake lower than the EAR. For the other nutrients investigated, the proportion below the EAR (AI) increased as the E-DII score increased in the order of T1, T2, T3. In the poorest dietary group, the T3 group, over 80% of participants had intake of vitamins A, D, E, B1, B2, B6, C, folate, calcium, magnesium, and iron that was below the EAR (AI).

### 3.4. Food Group Intake Profiles by Tertiles of NRF9.3 and E-DII Scores

The differences in the distribution of food group intake across tertiles of NRF9.3 score were investigated. The NRF9.3 scores increased with an increase in the intake of vegetables, fruits, fish and shellfish, legumes, potatoes, and seaweed. The NRF9.3 scores increased with a decrease in the intake of fats/oils and confectionery (Table 2).

Similarly, the differences in the distribution of food group intake across the tertiles of E-DII score were investigated. The E-DII scores increased with a decrease in the intake of vegetables, legumes, fruits, fish, and shellfish. The E-DII scores increased with an increase in the intake of wheat flour and wheat products (Table 3).

### 3.5. Sensitivity Analysis

In total, 17 (15.7%) participants were excluded from the sensitivity analysis because of under-reporting of EI. The main nutrients contributing to both dietary indices were not different after participant exclusion (Appendix A). However, the trend of food group intake across the tertiles was different: intake of vegetables, fruits, and potatoes increased as the NRF9.3 score increased (Table 4). A decrease in the intake of vegetables, fruits, and legumes and an increase in the intake of wheat flour and wheat products led to an increase in the E-DII score (Table 5). Considering all the results of Table 2, Table 3, Table 4 and Table 5, the food groups with commonly significant differences in intake among the tertiles of NRF9.3 and E-DII scores were vegetables and fruits.

## 4. Discussion

This study evaluated the maternal dietary quality and inflammatory potential of a cohort of pregnant Japanese women using the NRF9.3 and E-DII indices. The mean (SD) NRF9.3 and E-DII scores were 602 (106) and 1.00 (1.55), and the NRF9.3 and E-DII scores depicted a significant inverse correlation (Figure 2). Based on the tertile stratification of each index, the nutrients that had profound effects on the scores of each index were identified. In case of the NRF9.3 score, dietary fiber, iron, potassium, magnesium, and vitamin C contributed to the variation of total score across the tertiles (Figure 3). On the other hand, for the E-DII score, dietary fiber, vitamin A, niacin, vitamin E, β-carotene, magnesium, vitamin B1, vitamin C, zinc, vitamin B6, and folate contribute to the variation of the total score across the tertiles (Figure 4). The food groups whose intake were positively associated with dietary quality, as assessed using either NRF9.3 or E-DII, were vegetables and fruits even after adjustment for potential confounders and after considering the influence of misreporting EI. In addition, intake of wheat flour and wheat products was positively associated with inflammatory potential, as assessed using E-DII (Table 2, Table 3, Table 4 and Table 5).

The NRF9.3 score represents the adherence to the national RDV. To the best of our knowledge, this is the first study to apply the NRF9.3 index to assess the dietary quality of pregnant women. A closer look at the breakdown of the NRF9.3 and analysis of the component scores revealed the extent of insufficient intake of qualifying nutrients as well as the extent of excess intake of disqualifying nutrients, compared to the RDV (Figure 3). Among the qualifying nutrients, the component scores for iron were low in all the tertile groups. The component scores of vitamins A and D were much lower than 100 in the highest tertile, suggesting insufficient intake of these nutrients. Regarding the disqualifying nutrients, the differences between the three tertile groups were small. As shown in Appendix A, the participants in the lowest NRF9.3 tertile group (T1) were also more likely to have insufficient intake of other essential nutrients, such as folate, vitamins B1, B2, and B6; however, these nutrients are not components of the NRF9.3 score.

The DII score characterizes the inflammatory potential of a diet based on the literature-based evidence, which is globally applicable. In recent years, the DII of pregnant mothers has been reported to be associated with birth size and childhood health, and has received much attention. However, previous studies that have investigated the DII score of pregnant women have focused only on the score, and have failed to examine which nutrients contribute to the score [11,28,29,30,48]. Since the nutrients that contribute to the DII score may differ depending on the target population, it may be desirable to describe not only the total score but also the parameter-specific scores. As shown in Figure 4, the parameter-specific scores for dietary fiber, vitamins A, B1, B2, D, and niacin were positive in all the tertile groups. The parameter-specific scores of most anti-inflammatory nutrients increased as the E-DII scores increased, suggesting that the major cause for an increase in final E-DII score was the low intake of dietary fiber, vitamins, and minerals. The parameter-specific scores of pro-inflammatory nutrients were below 0 in all tertile groups, except for cholesterol. Interestingly, the intake of saturated fatty acid, total fat, and carbohydrates were not major factors in increased E-DII scores in our cohort.

As shown in Table 2 and Table 3, the food groups whose intake were positively associated with higher dietary quality in either NRF9.3 or E-DII were legumes, vegetables, fruits, fish, and shellfish. A previous study stated that a diet high in bread, confectioneries, and soft drinks and low in fish and vegetables during pregnancy might be associated with low birth weight [12]. Another study reported that adherence to vegetable dietary patterns may be associated with a lower risk of preeclampsia [15]. A recent review reported that a high consumption of vegetables and fruits is associated with lower risks of preterm birth and low birth weight [49]. These findings signify that a high intake of fruits and vegetables is characteristic of high dietary quality during pregnancy. Our nutrient-based dietary assessment identified that a high NRF9.3 score and a low E-DII score were associated with a high intake of vegetables and fruits, a finding which is consistent with the previous studies.

The purpose of this study was to examine the quality of the diet, through the intake of foods, not supplements. Nutrients in dietary supplements are not metabolized in the same way as those in foods, and this puts supplement users at risk of excess intake. Also, from the viewpoint of sustaining a healthy dietary habit, we decided to exclude intake from supplements when assessing dietary quality in this study. However, we were aware that pregnant women often take vitamins and minerals through dietary supplements. It would be a challenge in the future to determine accurate nutrient intake from dietary supplements and evaluate both food-derived and supplement-derived nutrients.

The main strength of our study is that it is the first time that the NRF9.3 index has been applied to assess the overall dietary quality of pregnant women. In particular, using the comprehensive database for sugars [36], we were able to deduce the added sugar intake and NRF9.3 scores, which is in contrast to the previous study that calculated the nutritional score without considering sugars [50]. Importantly, the anthropometric and overall energy/nutrient intake characteristics of the participants were almost identical to those of the pregnant women cohort in National Health and Nutrition Survey (NHNS 2015–2017) [35,51]; therefore, the results are generalizable to some extent, although our cohort was a single-center cohort from an urban area in Japan (*n* = 108). Additionally, we assessed the overall dietary quality and inflammatory potential using two globally established metrics, NRF9.3 and E-DII, and found there was a strong negative correlation between them. The analysis of the component score or the parameter-specific score clarified that the major nutrients contributing to both indices were similar: dietary fiber, magnesium, vitamin C, and vitamin A. This result is especially important because the most of the previous DII studies have not revealed which nutrients are associated with the final score.

This study has several limitations. First, three-day food dietary records were used for dietary assessment; however, the minimum number of days required for estimating an individual’s average EI is usually longer than 3 days [52]. Second, the possibility of seasonal variation, which might have introduced bias in the assessment of average dietary intake, was not considered. Third, self-reported dietary records have the potential disadvantage of under-reporting of the dietary intake [45,46,47]. Nevertheless, all analyses of nutritional score calculations and food intake were adjusted for EI. Furthermore, sensitivity analyses were performed by excluding the under-reporters.

## 5. Conclusions

Our assessment of the overall dietary quality and inflammatory potential of the maternal diet using both NRF9.3 and E-DII scores for pregnant women in Japan revealed that dietary fiber, vitamins, and minerals were the major nutrients influencing the dietary quality. Higher intake of vegetables and fruits was associated with increased dietary quality and lower inflammatory potential. Hence, promoting the intake of these food groups may benefit pregnant women and help them to achieve a healthy diet during pregnancy.

## Figures and Tables

**Figure 1 nutrients-13-02854-f001:**
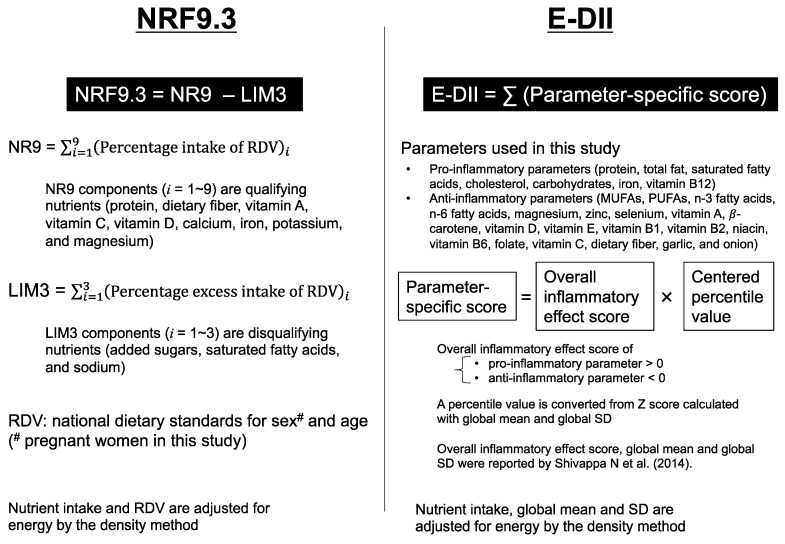
Summary of the dietary indices used in this study. The Nutrient-Rich Food Index 9.3 (NRF9.3) represents adherence to the national dietary intake standards [17,18]. The dietary Inflammatory Index (DII) is a comprehensive index of diet-derived inflammatory capacity, calculated using a common global mean and standard deviation (SD) deduced from the globally representative world database reported by Shivappa N et al. [19]. The energy adjusted DII (E-DII) is often used as an improved version of DII [20]. The details of these indices are described in the Materials and Methods section; NR, nutrient-rich; LIM, limiting nutrients; RDV, reference daily value; MUFAs, monounsaturated fatty acids; PUFAs, polyunsaturated fatty acids.

**Figure 2 nutrients-13-02854-f002:**
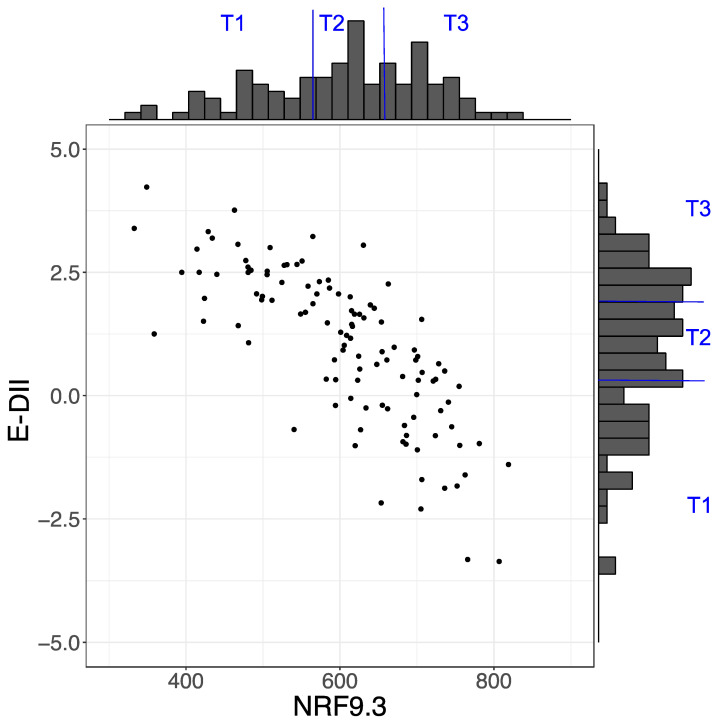
Distribution of Nutrient-Rich Food Index 9.3 (NRF9.3) and energy-adjusted dietary inflammatory index (E-DII) scores. Scatter plot showing the relationship between NRF9.3 and E-DII scores.

**Figure 3 nutrients-13-02854-f003:**
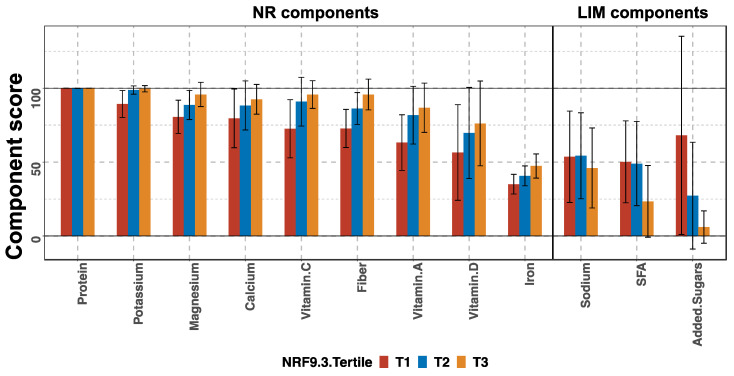
Component scores according to the tertiles (T) of total Nutrient-Rich Food Index 9.3 (NRF9.3) score in pregnant women of a birth cohort, BC-GENIST, in Japan. Bars show the mean and error bars represent standard deviation (SD). The nutrients constituting the nutrient-rich (NR) and limiting nutrient (LIM) sub-scores are shown from left to right, from the highest to lowest mean score of the entire cohort. For the NR sub-scores, a higher score indicates a higher dietary quality, while for the LIM sub-scores, a higher score indicates a lower dietary quality. SFA, saturated fatty acids. *n* = 36 (T1), 36 (T2), and 36 (T3).

**Figure 4 nutrients-13-02854-f004:**
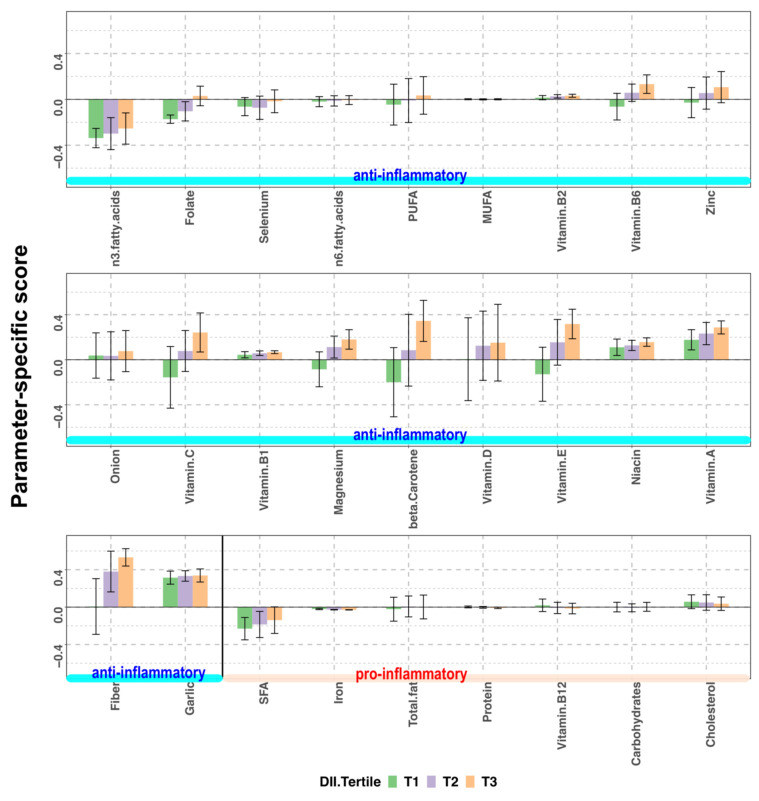
Parameter-specific scores according to the tertiles (T) of energy-adjusted dietary inflammatory index (E-DII) score in pregnant women of a birth cohort, BC-GENIST, in Japan. Bars represent the mean and error bars represent standard deviation (SD). The anti-inflammatory and pro-inflammatory parameters constituting the E-DII score are shown from left to right, from the lowest to the highest mean score of the entire cohort. Since the overall inflammatory effect scores of anti-inflammatory parameters are negative, lower intake of anti-inflammatory nutrients compared to the global mean result in a positive parameter-specific score. In contrast, as the overall inflammatory effect scores of pro-inflammatory parameters are positive, higher intake of pro-inflammatory nutrients result in a positive parameter-specific score. Higher E-DII scores indicate higher inflammatory potential. SFA, saturated fatty acids; MUFA, monounsaturated fatty acids; PUFA, polyunsaturated fatty acids; *n* = 36 (T1), 36 (T2), and 36 (T3).

**Table 1 nutrients-13-02854-t001:** Characteristics of the Japanese pregnant women across the tertile (T) categories of dietary index, BC-GENIST (2015–2019).

Characteristics	NRF9.3				E-DII			
	T1 (*n* = 36)	T2 (*n* = 36)	T3 (*n* = 36)	*p* Value ^a^	T1 (*n* = 36)	T2 (*n* = 36)	T3 (*n* = 36)	*p* Value ^a^
Maternal age (years)	32.6 ± 4.1	34.8 ± 4.3	34.2 ± 3.8	0.069	34.7 ± 4.0	33.8 ± 4.0	33.1 ± 4.4	0.258
Height (cm)	159.4 ± 5.2	158.6 ± 6.0	159.7 ± 5.3	0.673	160.2 ± 4.7	159.0 ± 6.6	158.5 ± 4.9	0.390
Pre-pregnancy weight (kg)	53.9 ± 6.8	54.0 ± 9.5	51.1 ± 7.7	0.223	52.3 ± 8.7	53.3 ± 9.1	53.4 ± 6.5	0.813
Pre-pregnancy BMI (kg/cm^2^)	21.3 ± 2.9	21.4 ± 2.8	20.0 ± 2.6	0.064	20.3 ± 3	21.0 ± 2.9	21.2 ± 2.5	0.350
Parity (multipara)	18 (50)	16 (44.4)	22 (61.1)	0.354	19 (52.8)	19 (52.8)	18 (50)	0.964
Energy intake (EI) (kcal/day) ^b^	1690 ± 280	1651 ± 271	1698 ± 286	0.747	1714 ± 303	1648 ± 247	1677 ± 283	0.607
Smoking in pregnancy	0 (0)	0 (0)	0 (0)	−	0 (0)	0 (0)	0 (0)	−
Maternal educational attainment, university or higher degree	23 (63.9)	25 (69.4)	26 (72.2)	0.741	26 (72.2)	28 (77.8)	20 (55.6)	0.107
Household income (≥ 6 million yen per year)	24 (68.6)	25 (69.4)	25 (69.4)	0.996	24 (66.7)	29 (80.6)	21 (60.0)	0.159
Fetal sex, male	21 (58.3)	13 (36.1)	18 (50)	0.163	19 (52.8)	16 (44.4)	17 (47.2)	0.771

Values are represented as mean ± standard deviation (SD) or *n* (%). ^a^ Differences among tertiles for each nutritional score were tested with one-way ANOVA for continuous variables and Chi-square test for categorical variables. ^b^ The individual energy intake was estimated by the mean values over three days. NRF9.3, Nutrient-Rich Food Index 9.3. E-DII, energy-adjusted dietary inflammatory index. BMI, body mass index.

**Table 2 nutrients-13-02854-t002:** Food group intake across the tertile (T) categories of Nutrient-Rich Food Index 9.3 (NRF9.3) score in pregnant women of a birth cohort, BC-GENIST, in Japan.

Food Group (g)	T1 (*n* = 36)	T2 (*n* = 36)	T3 (*n* = 36)	*p* for Trend ^a^
Rice and Rice products	131.5 ± 67.7	124.1 ± 42.6	146.6 ± 61.2	0.44
Wheat flour and Wheat products	76.6 ± 46.7	85.8 ± 43	73.7 ± 53.6	0.93
Potatoes	11.8 ± 13.2	18.1 ± 15.9	22.0 ± 20.3	0.015
Legumes	16.7 ± 15.5	23.6 ± 24.8	33.9 ± 30.7	0.0062
Seeds and Nuts	0.7 ± 1.3	1.0 ± 1.9	0.9 ± 1.6	0.79
Vegetables	101.3 ± 46.6	135.1 ± 56.5	167.5 ± 69.8	<0.0001
Fruits	25.5 ± 26.9	35.4 ± 32.3	59.6 ± 52.7	0.00011
Mushrooms	5.2 ± 5.6	6.4 ± 8.4	6.4 ± 7.0	0.59
Seaweeds	3.9 ± 4.7	7.7 ± 9.1	7.1 ± 7.8	0.038
Fish and Shellfish	8.8 ± 10.0	18.7 ± 20.9	22.8 ± 20.6	0.00077
Meat and Poultry	57.9 ± 25.5	55.3 ± 26.2	51.9 ± 26.4	0.35
Egg	15.6 ± 13.0	22.1 ± 15.1	20.2 ± 12.7	0.078
Milk and Dairy Products	95.4 ± 86.6	95.4 ± 68.3	82.5 ± 65.1	0.49
Fats and oils	6.1 ± 2.9	6.4 ± 3.5	4.7 ± 2.3	0.021
Confectionery	25.9 ± 28.4	19.2 ± 19.5	16.8 ± 16.1	0.044
Sugar-sweetened beverages	54.4 ± 71.9	43.7 ± 66.4	26.4 ± 30.0	0.051
Seasonings and Spices	30.0 ± 13.0	28.0 ± 9.6	28.2 ± 10.2	0.90

Food group intake is expressed as intake/1000 kcal. Values (g) are represented as mean ± standard deviation (SD). ^a^ For statistical analysis, linear regression model was used for testing a trend, adjusted for maternal age, pre-pregnancy body mass index (BMI), and educational attainment.

**Table 3 nutrients-13-02854-t003:** Food group intake across the tertile (T) categories of energy-adjusted dietary inflammatory index (E-DII) in pregnant women of a birth cohort, BC-GENIST, in Japan.

Food Group (g)	T1 (*n* = 36)	T2 (*n* = 36)	T3 (*n* = 36)	*p* for Trend ^a^
Rice and Rice products	133.3 ± 59.7	130.9 ± 50.1	137.9 ± 65.7	0.53
Wheat flour and Wheat products	61.8 ± 44.1	88.6 ± 46.9	85.8 ± 49.0	0.039
Potatoes	20.5 ± 19.9	17.3 ± 17.7	14.1 ± 12.7	0.17
Legumes	40.6 ± 33.9	17.8 ± 14.3	15.8 ± 14.5	<0.0001
Seeds and Nuts	1.0 ± 1.9	0.8 ± 1.5	0.7 ± 1.3	0.76
Vegetables	178.5 ± 60	141.3 ± 54.1	84.1 ± 35.8	<0.0001
Fruits	55.8 ± 55.6	33.0 ± 26.4	31.7 ± 31.8	0.0067
Mushrooms	6.9 ± 8.3	6.0 ± 7.0	5.1 ± 5.7	0.38
Seaweeds	7.5 ± 8.5	6.4 ± 7.2	4.8 ± 6.8	0.058
Fish and Shellfish	24.0 ± 25.2	14.4 ± 14	11.9 ± 12.3	0.0068
Meat and Poultry	53.8 ± 25.5	58.3 ± 27.9	53.0 ± 24.6	0.88
Egg	20.9 ± 14	20.9 ± 15.1	16.1 ± 11.8	0.069
Milk and Dairy Products	89.6 ± 73.4	83.4 ± 67.7	100.2 ± 80.0	0.53
Fats and oils	5.3 ± 2.6	6.2 ± 3.6	5.7 ± 2.7	0.43
Confectionery	19.8 ± 19.7	19.3 ± 20.1	22.9 ± 26.3	0.45
Sugar-sweetened beverages	37.4 ± 58.6	30.7 ± 47.0	56.3 ± 70.0	0.16
Seasonings and Spices	30.3 ± 10.7	29.0 ± 10.8	26.9 ± 11.4	0.061

Food group intake is expressed as intake/1000 kcal. Values (g) are represented as mean ± standard deviation (SD). ^a^ For statistical analysis, linear regression model was used to test for trends, adjusted for maternal age, pre-pregnancy body mass index (BMI), and educational attainment.

**Table 4 nutrients-13-02854-t004:** Sensitivity analysis for food group intake across the tertile (T) categories of Nutrient-Rich Food Index 9.3 (NRF9.3) score after excluding the participants with misreported energy intake.

Food Group (g)	T1 (*n* = 30)	T2 (*n* = 30)	T3 (*n* = 31)	*p* for Trend ^a^
Rice and Rice products	132.7 ± 68.0	122.8 ± 44.3	152.2 ± 61.9	0.30
Wheat flour and Wheat products	69.1 ± 45.6	85.8 ± 44.5	66.5 ± 49.5	0.79
Potatoes	10.3 ± 10.2	18.9 ± 15.9	21.9 ± 21.5	0.015
Legumes	16.5 ± 14.2	25.1 ± 26.4	30.7 ± 30.0	0.11
Seeds and Nuts	0.7 ± 1.3	1.2 ± 2.0	0.9 ± 1.6	0.96
Vegetables	110.0 ± 44.5	130.9 ± 46.8	163.4 ± 63.7	0.00044
Fruits	22.1 ± 22.7	35.1 ± 31.9	60.4 ± 46.9	<0.0001
Mushrooms	6.0 ± 5.9	6.4 ± 8.9	7.0 ± 7.2	0.59
Seaweeds	3.3 ± 3.4	7.6 ± 9.4	5.8 ± 6.2	0.33
Fish and Shellfish	9.2 ± 10.7	20.3 ± 21.9	18.6 ± 15.5	0.14
Meat and Poultry	61.0 ± 24.6	51.7 ± 24.7	55.7 ± 25.8	0.57
Egg	14.8 ± 13.4	22.0 ± 15.4	19.3 ± 12.2	0.14
Milk and Dairy Products	94.3 ± 90.1	89.5 ± 70.3	85.1 ± 67.9	0.73
Fats and oils	5.6 ± 2.6	6.3 ± 3.6	4.6 ± 2.4	0.10
Confectionery	25.2 ± 26.5	21.0 ± 20.3	18.4 ± 15.8	0.13
Sugar-sweetened beverages	60.6 ± 76.2	43.6 ± 66.6	28.9 ± 32.9	0.088
Seasonings and Spices	30.3 ± 12.3	28.3 ± 10.2	27.0 ± 10.8	0.43

Food group intake is expressed as intake/1000 kcal. Values (g) are represented as mean ± standard deviation (SD). ^a^ For statistical analysis, linear regression model was used for testing a trend, adjusted for maternal age, pre-pregnancy body mass index (BMI), and educational attainment.

**Table 5 nutrients-13-02854-t005:** Sensitivity analysis for food group intake across the tertile (T) categories of energy-adjusted dietary inflammatory index (E-DII) score after excluding the participants with misreported energy intake.

Food Group (g)	T1 (*n* = 30)	T2 (*n* = 30)	T3 (*n* = 31)	*p* for Trend ^a^
Rice and Rice products	139.5 ± 61.5	125.4 ± 52.5	143.0 ± 64.4	0.66
Wheat flour and Wheat products	53.3 ± 42.0	86.7 ± 43.9	80.8 ± 48.8	0.0090
Potatoes	22.1 ± 20.5	15.8 ± 18.6	13.6 ± 9.9	0.12
Legumes	39.0 ± 34.9	17.5 ± 14.6	16.2 ± 12.7	0.0015
Seeds and Nuts	1.2 ± 2.1	0.9 ± 1.5	0.8 ± 1.4	0.74
Vegetables	174.7 ± 53.7	140.2 ± 44.6	91.8 ± 36.4	<0.0001
Fruits	55.9 ± 49.9	34.2 ± 27.5	28.6 ± 30.0	0.0080
Mushrooms	7.5 ± 8.8	6.1 ± 7.4	5.8 ± 5.8	0.34
Seaweeds	6.3 ± 7.2	5.8 ± 7.3	4.7 ± 6.4	0.57
Fish and Shellfish	21.3 ± 22.5	14.4 ± 13.9	12.7 ± 12.9	0.28
Meat and Poultry	53.5 ± 24.5	59.3 ± 27.2	55.6 ± 24.0	0.95
Egg	19.6 ± 13.5	21.2 ± 16.0	15.5 ± 11.9	0.13
Milk and Dairy Products	93.4 ± 78	77.3 ± 66.9	97.8 ± 82.7	0.77
Fats and oils	5.2 ± 2.7	6.3 ± 3.7	5.0 ± 2.1	0.92
Confectionery	21.2 ± 19.7	22.8 ± 20.9	20.6 ± 23.5	0.98
Sugar-sweetened beverages	40.9 ± 62.4	34.7 ± 50.4	56.5 ± 70.8	0.40
Seasonings and Spices	29.6 ± 11.5	29.3 ± 11.7	26.7 ± 10.1	0.12

Food group intake is expressed as intake/1000 kcal. Values (g) are represented as mean ± standard deviation (SD). ^a^ For statistical analysis, linear regression model was used for testing a trend, adjusted for maternal age, pre-pregnancy body mass index (BMI), and educational attainment.

## Data Availability

The datasets used and/or analyzed during the current study are shown in Appendix A.

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
