# Peer review of "Application of the Nutrient-Rich Food Index 9.3 and the Dietary Inflammatory Index for Assessing Maternal Dietary Quality in Japan: A Single-Center Birth Cohort Study"

_nutrients, 2021, doi:10.3390/nu13082854_

Round 1

Reviewer 1 Report

GENERAL COMMENT

The authors aimed to assess the overall dietary quality and inflammatory potential of pregnant women during mid-gestation using two indices, namely the Nutrient-Rich Food Index 9.3 and the Dietary Inflammatory Index. The authors used data from the BC-GENIST Japanese study. Overall, the authors found an inverse correlation between the scores. They were also able to show that dietary fiber, vitamin C, vitamin A, and magnesium commonly contributed to the variability of both indices and that vegetables and fruits are the food groups mainly associated with high dietary quality scores among pregnant Japanese women. This paper is very interesting and raises important concerns regarding the way diet quality is viewed and investigated by the research community during pregnancy.

MAJOR COMMENTS

  1. Could the authors specify why they chose to use those dietary indices as they do not appear to have been validated for Japanese pregnant women populations? Maybe adding a sentence or two on why those were chosen, and others were not.
  2. I think a summary of the indices used should be presented in one Figure. Some information about the description of the indices should be removed from the introduction.
  3. For the adjustment of dietary variables, why did the authors not adjust the results for the socioeconomic status (household income) of the participants, knowing that this is also a parameter that can have an influence on nutritional intakes?
  4. The abbreviations DII and E-DII are used interchangeably in some sections and in the figures. I would suggest standardizing for E-DII as every data presented is adjusted for energy intake.
  5. Is the DII score really a measure of food quality? I'm not sure. Perhaps the term food quality should be used with caution.
  6. Could the authors specify why they chose to use the NHNS cohort to confirm their results? Does it include a pregnancy cohort? I did not find the explanation for that decision and this clarification anywhere in the manuscript. A section should be added about the comparison with the NHNS population and to clarify the use of those data in the Materials and Methods section.
  7. I would suggest clarifying the Dietary data section of the Materials and Methods section regarding the completion of the food diaries and the use for the photobook. There is a need to describe more in depth the process and the tools used (photobook).
  8. When the authors discuss about the contribution of some nutrients, foods or food groups to the score or their effect on the score. I suggest that they replace it by the contribution of some nutrients, foods or food groups to the variation of the total score between groups or their effects on the variation of the total score between groups.
  9. Discussion sections should be added: limits of not including the supplement use in the analyses; discussing limits and strengths of the indices used and if their use is appropriate for the gestational context.
  10. A comprehensive review to improve the quality of the use of English, especially for certain wording, should be done

MINOR COMMENTS

  1. Abstract (line 36) I suggest adding ‘Consistent with previous studies using dietary pattern analysis, this study showed that the food groups mainly associated with high dietary quality and low inflammatory potential among pregnant Japanese women were vegetables and fruits’
  2. Introduction (lines 55-58). I suggest reformulation ‘However, these studies have mainly focused on certain foods or food groups. Moreover, studies involving comprehensive dietary quality assessments using multiple nutrients, rather than an individual nutrient, are less common’.
  3. Introduction (lines 59-83). Description of the indices should be shortened and detailed in the Methods section.
  4. Introduction (line 70). The authors introduce the influence of skipping breakfast on diet quality. However, it is not discussed elsewhere in the manuscript and their data doesn’t include the meal frequency. ‘such as skipping breakfast’ should then be removed.
  5. Introduction (lines 83-84). I suggest reformulation: ‘five nutrients (dietary fiber, vitamin A, vitamin C, vitamin D, and magnesium) are included as anti-inflammatory nutrients in the DII index.’
  6. Materials and Methods (lines 136-142). Does the photo book was used to correct the dietary record reported by the participants? This section lacks clarity.
  7. Materials and Methods (lines 196-198) I suggest reformulationAll nutrient data in the dietary records and the global dietary intake database were converted to values per 1,000 kcal by dividing these data by the EI from the diet and multiplied by 1,000.’
  8. Results (line 254). ‘There was a strong inverse correlation between the NRF9.3 scores and the E-DII scores (r = −0.793, p < 2.2 × 10−6)’. Do the authors mean the total score of each index, or components’ scores of each index?
  1. Discussion (line 374). The abbreviation DRV is used without being previously specified.
  2. Conclusions (line 440). I suggest adding ‘Higher intakes of vegetables and fruits were associated with increased dietary quality and lower inflammatory potential
  3. Tables 1-2-3-S4-S5-S6-S7 (titles). I would standardize the titles with the formulation ‘across tertile (T) categories
  4. Figure 2 (Y axis). Component
  1. The term ‘carbohydrate’ should be used in its plural form (carbohydrates) in the text, tables, and figures.
  2. The expressions ‘food group(s)’ and ‘intake(s)’ are used in their singular and plural forms in the text, tables, and figures. I suggest standardizing to their plural forms where appropriate.

Author Response

Response to Reviewer 1 Comments

Point 1: GENERAL COMMENT

The authors aimed to assess the overall dietary quality and inflammatory potential of pregnant women during mid-gestation using two indices, namely the Nutrient-Rich Food Index 9.3 and the Dietary Inflammatory Index. The authors used data from the BC-GENIST Japanese study. Overall, the authors found an inverse correlation between the scores. They were also able to show that dietary fiber, vitamin C, vitamin A, and magnesium commonly contributed to the variability of both indices and that vegetables and fruits are the food groups mainly associated with high dietary quality scores among pregnant Japanese women. This paper is very interesting and raises important concerns regarding the way diet quality is viewed and investigated by the research community during pregnancy.

Response 1: Thank you for your time spent reviewing our manuscript and valuable comments and suggestions.  

MAJOR COMMENTS

Point 2: 1. Could the authors specify why they chose to use those dietary indices as they do not appear to have been validated for Japanese pregnant women populations? Maybe adding a sentence or two on why those were chosen, and others were not.

Response 2: Thank you for your comments. We agree that previous studies on Japanese pregnant women have applied dietary pattern analysis such as the paper by Okubo H et al (PMID: 21929833).

The authors believe that both food-based and nutrient-based dietary indices are useful in assessing the overall quality and properties of a pregnant woman's diet. However, the international use of food-based indices will require harmonization of the food database because foods frequently consumed are unique to each country's dietary culture. In contrast, nutrients are universal so that nutrient-based dietary indices are globally used without further processing. Therefore, in this study, we used two dietary indices that are commonly used in the world, which cover a wide range of nutrients.

The importance of maternal diet has been recently reaffirmed, based on Developmental Origin of Health and Disease (DOHaD) theory, for prevention of future non-communicable diseases of offspring in addition to supporting healthy pregnancy and delivery. From this point of view, it can be said that there are no completely validated dietary indices for pregnant women (and their children), not only in Japan but also internationally.

We have added the following sentences in the text (page 2, lines 60-68)

“Both food-based and nutrient-based dietary indices are useful in assessing the overall quality and/or properties of a pregnant woman's diet. However, the international use of food-based indices will require harmonization of the food database because foods frequently consumed are unique to each country's dietary culture. In contrast, nutrients are universal so that nutrient-based dietary indices can be globally used without further processing. Therefore, in this study, we used two dietary indices that are commonly used in the world, which cover a wide range of nutrients: the Nutrient-Rich Food index 9.3 (NRF9.3) and the dietary inflammatory index (DII).”

Point 3: 2. I think a summary of the indices used should be presented in one Figure. Some information about the description of the indices should be removed from the introduction.

Response 3: Thank you for your suggestions. In the revised manuscript, the summary of two dietary indices is presented in Figure 1 (page 2). A part of the information about the description of the indices was removed from the introduction.

Point 4: 3. For the adjustment of dietary variables, why did the authors not adjust the results for the socioeconomic status (household income) of the participants, knowing that this is also a parameter that can have an influence on nutritional intakes?

Response 4: Thank you for your suggestions. Preliminary analysis showed that household income was not related to nutritional intakes in our cohort. A previous study in Japanese pregnant women also reported the similar results (Murakami, K.; et al. Education, but not occupation or household income, is positively related to favorable dietary intake patterns in pregnant Japanese women: the Osaka Maternal and Child Health Study. Nutr Res 2009, 29, 164-172). We added the following sentences in the method section (page 7, lines 279-282)

“Household income was a potential confounder, but in our cohort, household income was not associated with dietary indices or food intake analyzed. A previous study on a Japanese population also reported that household income was not associated with any nutrients or foods examined [47]. Therefore, household income was not included as an adjustment factor in this analysis.”

Point 5: 4. The abbreviations DII and E-DII are used interchangeably in some sections and in the figures. I would suggest standardizing for E-DII as every data presented is adjusted for energy intake.

Response 5: Thank you for your comment. We carefully amended this point.

Point 6: 5. Is the DII score really a measure of food quality? I'm not sure. Perhaps the term food quality should be used with caution.

Response 6: We apologize for the confusion. We do not mean that the DII score is a measure of food quality. Instead, the DII score can be a dietary quality index, as expressed in several papers including PMID: 32800153.

              The dietary quality index is an index that quantifies and measures the degree of adequacy between actual intake of nutrients and/or foods and the reference intake, which is established based on scientific evidence assuring an optimal state of health. Since diets are composed of multiple components, not a single nutrient or a single food, the dietary quality index is used to assess them comprehensively. According to this definition, both NRF9.3 and DII can be said to be dietary quality indices. DII does measure how nutrient and food intake deviates from global standards. However, DII was originally developed as an indicator of inflammatory potential. Therefore, to avoid confusion, when referring to the DII score, a dietary index is used, and “quality” was removed from “dietary quality index”.

Point 7: 6. Could the authors specify why they chose to use the NHNS cohort to confirm their results? Does it include a pregnancy cohort? I did not find the explanation for that decision and this clarification anywhere in the manuscript. A section should be added about the comparison with the NHNS population and to clarify the use of those data in the Materials and Methods section.

Response 7: Thank you for pointing this out. Regarding the NHNS data, the mention of a pregnant population was missing. We have corrected this point and added an explanation in the Materials and Methods section (page 6, lines 260-265).

Point 8: 7. I would suggest clarifying the Dietary data section of the Materials and Methods section regarding the completion of the food diaries and the use for the photobook. There is a need to describe more in depth the process and the tools used (photobook).

Response 8: Thank for your comments. We revised the description in the Dietary data section as follows (page 4, lines 158-171).

“The participants were advised to avoid recording intakes of weekends and holidays. Approximately four weeks later, the participants returned the notebook to be checked by the study dietician. The dieticians used a photobook of 122 commonly eaten foods and dishes [34] to identify the portion sizes of the foods and beverages consumed. The pictures in this photobook were real-sized, accompanied by weight (g) of the food. Upon collection of the food dietary records, all the recorded sheets were checked by dietitians to clarify any ambiguous points, such as adding seasonings at the table, consumption of tea or other beverages, and snacking between meals. The dieticians then converted the portions of the foods consumed into estimated intakes (g). When a raw food such as vegetables, meat, fish, or egg was heat-cooked, the type of cooking was also checked and recorded. Each food item was coded according to the food numbers in the Standard Tables of Food Composition (STFC) in Japan, 2015 [35], so that energy and nutrient intakes could be calculated.”

Point 9: 8. When the authors discuss about the contribution of some nutrients, foods or food groups to the score or their effect on the score. I suggest that they replace it by the contribution of some nutrients, foods or food groups to the variation of the total score between groups or their effects on the variation of the total score between groups.

Response 9: Thank you for your suggestions. We corrected it (page 14, lines 472-476).

“In case of the NRF9.3 score, dietary fiber, iron, potassium, magnesium, and vitamin C contributed to the variation of total score across the tertiles (Figure 3). On the other hand, for the E-DII score, dietary fiber, vitamin A, niacin, vitamin E, b-carotene, magnesium, vitamin B1, vitamin C, zinc, vitamin B6, and folate contribute to the variation of the total score across the tertiles (Figure 4).”

Point 10:             9. Discussion sections should be added: limits of not including the supplement use in the analyses; discussing limits and strengths of the indices used and if their use is appropriate for the gestational context.

Response 10: Thank you for your important comments. The purpose of this study was to examine the quality of the diet, through the intake of foods, not supplements. Nutrients in dietary supplements are not metabolized in the same way as those in foods, and this puts supplement users at risk of excess intakes. Also, from the viewpoint of sustaining a healthy dietary habit, we decided to exclude intakes from supplements when assessing dietary quality in this study. However, we were aware that nutrient intakes from dietary supplements were not ignorable for pregnant women. We experienced that it is not easy to accurately identify the amount of nutrients taken from dietary supplements. In the future, it would be necessary to develop methods to evaluate nutrients from both foods and supplements. Therefore, I added the following paragraph to the Discussion section (page 15, lines 521-529).

“The purpose of this study was to examine the quality of the diet, through the intake of foods, not supplements. Nutrients in dietary supplements are not metabolized in the same way as those in foods, and this puts supplement users at risk of excess intakes. Also, from the viewpoint of sustaining a healthy dietary habit, we decided to exclude intakes from supplements when assessing dietary quality in this study. However, we were aware that pregnant women often take vitamins and minerals through dietary supplements. It would be a challenge in the future to determine accurate nutrient intakes from dietary supplements and evaluate both food-derived and supplement-derived nutrients.”

Point 11:             10. A comprehensive review to improve the quality of the use of English, especially for certain wording, should be done.

Response 11: We have it checked by an English proofreading expert and corrected.

MINOR COMMENTS

Point 12:             1. Abstract (line 36) I suggest adding ‘Consistent with previous studies using dietary pattern analysis, this study showed that the food groups mainly associated with high dietary quality and low inflammatory potential among pregnant Japanese women were vegetables and fruits’

Response 12: We have corrected it (page 1, lines 35-38)

“Consistent with the previous studies that used dietary pattern analysis, this study also demonstrated that vegetables and fruits were the food groups chiefly associated with high dietary quality and low inflammatory potential among pregnant Japanese women.”

Point 13:             2. Introduction (lines 55-58). I suggest reformulation ‘However, these studies have mainly focused on certain foods or food groups. Moreover, studies involving comprehensive dietary quality assessments using multiple nutrients, rather than an individual nutrient, are less common’.

Response 13: We have corrected it (page 2, lines 56-59)

“However, these studies have mainly focused on certain foods or food groups. Moreover, studies involving comprehensive dietary quality assessments using multiple nutrients, rather than an individual nutrient, are less common.”

Point 14:             3. Introduction (lines 59-83). Description of the indices should be shortened and detailed in the Methods section.

Response 14: Thank you for your suggestion. We removed the details from the Introduction section and moved them to the Materials and Methods section.

Point 15:             4. Introduction (line 70). The authors introduce the influence of skipping breakfast on diet quality. However, it is not discussed elsewhere in the manuscript and their data doesn’t include the meal frequency. ‘such as skipping breakfast’ should then be removed.

Response 15: Thank you for your comment. We removed these citations.

Point 16:             5. Introduction (lines 83-84). I suggest reformulation: ‘five nutrients (dietary fiber, vitamin A, vitamin C, vitamin D, and magnesium) are included as anti-inflammatory nutrients in the DII index.’

Response 16: We have corrected it (page 6, lines 228-230)

“Among the nine NRF9.3 qualifying nutrients, five nutrients (dietary fiber, vitamin A, vitamin C, vitamin D, and magnesium) are included as anti-inflammatory nutrients in the DII index. “

Point 17:             6. Materials and Methods (lines 136-142). Does the photo book was used to correct the dietary record reported by the participants? This section lacks clarity.

Response 17: As described in Response 8, the study dietitian used the photobook to convert the portion size of foods and beverages consumed and reported by the participants to their weight (g).

Point 18:             7. Materials and Methods (lines 196-198) I suggest reformulation ‘All nutrient data in the dietary records and the global dietary intake database were converted to values per 1,000 kcal by dividing these data by the EI from the diet and multiplied by 1,000.’

Response 18: We have corrected it (page 6, lines 245-247).

“All nutrient data in the dietary records and the global dietary intake database were converted to values per 1,000 kcal by dividing these data by the EI from the diet and multiplied by 1,000.”

Point 19:             8. Results (line 254). ‘There was a strong inverse correlation between the NRF9.3 scores and the E-DII scores (r = −0.793, p < 2.2 × 10−6)’. Do the authors mean the total score of each index, or components’ scores of each index?

Response 19: We mean the total score of each index. Therefore, we added “total” to that sentence (page 8, line 335-336).

“There was a strong inverse correlation between the NRF9.3 total scores and the E-DII total scores (r = −0.793, p < 2.2 × 10−6).”

Point 20:            9. Discussion (line 374). The abbreviation DRV is used without being previously specified.

Response 20: Thank you for pointing out this error. We have corrected it (page 14, lines 482-487).

“The NRF9.3 score represents the adherence to the national RDV. To the best of our knowledge, this is the first study to apply the NRF9.3 index to assess the dietary quality of pregnant women. A closer look at the breakdown of the NRF9.3 and analysis of the component scores revealed the extent of insufficient intake of qualifying nutrients as well as the extent of excess intake of disqualifying nutrients, compared to the RDV (Figure 3).”

Point 21:            10.Conclusions (line 440). I suggest adding ‘Higher intakes of vegetables and fruits were associated with increased dietary quality and lower inflammatory potential’

Response 21: Thank you for your suggestion. We have corrected it (page 16, lines 558-561).

“Higher intake of vegetables and fruits was associated with increased dietary quality and lower inflammatory potential.”

Point 22:            11. Tables 1-2-3-S4-S5-S6-S7 (titles). I would standardize the titles with the formulation ‘across tertile (T) categories’

Response 22: Thank you for the point about the standardization of the titles. We have corrected them.

This is not something that was pointed out by the reviewers, but please allow us to make the following changes to improve this manuscript. There is a " Intake of vegetables and fruits was positively associated with high NRF9.3 scores and negatively associated with high E-DII scores ~" in the results mentioned in the abstract of this paper, which was derived from both the main analysis and the sensitivity analysis (Tables S6 and S7 in the old manuscript). We have moved the old Tables S6 and S7 to the Results section as Tables 4 and 5 because the content of the abstract should be included in the main text, not in the supplementary material.

Point 23:            12. Figure 2 (Y axis). Component

Response 23: Thank you for pointing out this error. We have corrected it.

Point 24:            1. The term ‘carbohydrate’ should be used in its plural form (carbohydrates) in the text, tables, and figures.

Response 24: Thank you for pointing out this error. We have corrected it.

Point 25:            2. The expressions ‘food group(s)’ and ‘intake(s)’ are used in their singular and plural forms in the text, tables, and figures. I suggest standardizing to their plural forms where appropriate.

Response 25: Thank you for your comments. We have corrected it where appropriate. Regarding “intake”, the plural form will also be intake in general (https://www.wordhippo.com/what-is/the-plural-of/intake.html). An example sentence is “You should limit your daily intake of fats and sugars.” (https://www.merriam-webster.com/dictionary/intake)

Reviewer 2 Report

This study applied the Nutrient-Rich Food Index 9.3 and the Dietary Inflammation Index to assess maternal dietary quality in Japan. While mainly a descriptive study, I found it quite informative and the manuscript is clear and well-written. I have some questions/suggestions that I hope can further improve the manuscript:

-Please discuss the advantage(s) of nutrient-based indices over food-based indices to justify the need for this research.

-Line 124: Do the 3-d food diaries used include both weekday and weekend day?

-Line 152: It was stated that “In this study, dietary intake from supplements was not considered as our intention was the assessment of dietary quality from foods and beverages.” However, supplements likely have great influence on nutrients, especially micronutrients, intakes, and the authors should thus discuss potential limitations of excluding this information. For example, iron supplements are pretty common during pregnancy, and would have substantially reduced the proportions of participants below EAR (line 287).

-Line 162-169: It seems that different standards (RDA, AI, DG, and WHO) have been used for the scoring of different nutrients within the NRF9.3. Please discuss suitability, appropriateness, and implication of this approach.

-Section 2.4 and 2.5: Since the authors have stated the potential maximum score for NRF9.3, I wonder if it is useful to also state the theoretical minimum. For E-DII, could the authors also state the plausible range?

-Line 208: Besides under-reporter, is there any over-reporter?

-Line 300-309: I am quite confused by this paragraph and the figure, maybe because of the way DII is constructed (higher score is worse), thus making interpretations difficult. I wonder if the authors can think of a better way to present these data. If not, maybe it will help the readers by providing examples (e.g. referring to a specific component score when discussing differences by tertiles).

-Line 332: Is it possible to distinguish between good and bad oils/fats?

-Line 349: This sensitivity analysis must be described in Methods too to prepare the readers.

-Table S4: The categorization of nutrients into energy-producing, vitamins, and minerals is not clear.

-Line 413: It was mentioned that “The main strength of our study is that it is the first time that the NRF9.3 index been applied to assess the overall dietary quality of pregnant women.” But why is this important? Is NRF9.3 better than other diet quality index, and if so how so?

-Line 430: It was mentioned that seasonal variation was not considered, but I think you have the data as inferred from the Methods, so why not adjust for seasonality in your analyses?

Author Response

Response to Reviewer 2 Comments

Point 1: This study applied the Nutrient-Rich Food Index 9.3 and the Dietary Inflammation Index to assess maternal dietary quality in Japan. While mainly a descriptive study, I found it quite informative and the manuscript is clear and well-written. I have some questions/suggestions that I hope can further improve the manuscript: 

Response 1: Thank you for your time spent reviewing our manuscript and valuable comments and suggestions. 

Point 2: Please discuss the advantage(s) of nutrient-based indices over food-based indices to justify the need for this research.

Response 2: Thank you for your comments. We added the following sentences to justify the research using nutrient-based indices (page 2, lines 60-68).

“Both food-based and nutrient-based dietary indices are useful in assessing the overall quality and/or properties of a pregnant woman's diet. However, the international use of food-based indices will require harmonization of the food database because foods frequently consumed are unique to each country's dietary culture. In contrast, nutrients are universal so that nutrient-based dietary indices are globally used without further processing. Therefore, in this study, we used two dietary indices that are commonly used in the world, which cover a wide range of nutrients: the Nutrient-Rich Food Index 9.3 (NRF9.3) and the energy-adjusted dietary inflammatory index (E-DII) (Figure 1)”

Point 3:Line 124: Do the 3-d food diaries used include both weekday and weekend day?

Response 3: The participants were advised to avoid recording intakes of weekends and holidays to investigate the usual intake of nutrients and foods. We added the following description to the revised manuscript (page 4, lines 158-159).

“The participants were advised to avoid recording intakes of weekends and holidays.”

Point 4:-Line 152: It was stated that “In this study, dietary intake from supplements was not considered as our intention was the assessment of dietary quality from foods and beverages.” However, supplements likely have great influence on nutrients, especially micronutrients, intakes, and the authors should thus discuss potential limitations of excluding this information. For example, iron supplements are pretty common during pregnancy, and would have substantially reduced the proportions of participants below EAR (line 287).

Response 4: Thank you for your important comments. The purpose of this study was to examine the quality of the diet, through the intake of foods, not supplements. Nutrients in dietary supplements are not metabolized in the same way as those in foods, and this puts supplement users at risk of excess intakes. Also, from the viewpoint of sustaining a healthy dietary habit, we decided to exclude intakes from supplements when assessing dietary quality in this study. However, we were aware that nutrient intakes from dietary supplements were not ignorable for pregnant women. In the future, it would be necessary to develop methods to evaluate nutrients from both foods and supplements. Therefore, we added the following paragraph to the Discussion section (page 15, lines 521-529).

“The purpose of this study was to examine the quality of the diet, through the in-take of foods, not supplements. Nutrients in dietary supplements are not metabolized in the same way as those in foods, and this puts supplement users at risk of excess intake. Also, from the viewpoint of sustaining a healthy dietary habit, we decided to exclude intake from supplements when assessing dietary quality in this study. However, we were aware that pregnant women often take vitamins and minerals through dietary supplements. It would be a challenge in the future to determine accurate nutrient intake from dietary supplements and evaluate both food-derived and supplement-derived nutrients.”

Point 5:-Line 162-169: It seems that different standards (RDA, AI, DG, and WHO) have been used for the scoring of different nutrients within the NRF9.3. Please discuss suitability, appropriateness, and implication of this approach.

Response 5: The reason for using different standards is due to the differences in the standards presented in the DRI. For added sugars, we followed the WHO recommendations because, as noted in the text, there are no standards in Japan. For nutrients other than added sugars, the text has been slightly modified for clarification (page 5, lines 192-200).

“For six nutrients (protein, vitamins A and C, calcium, iron, and magnesium), the recommended dietary allowance (RDA) was indicated in the DRIs, the RDA was used as the RDV for those nutrients. For vitamin D, adequate intake (AI) was used as RDV, because RDA for vitamin D was not provided in DRIs. For dietary fiber, potassium, saturated fatty acids, and sodium, the tentative dietary goal for preventing life-style-related diseases (DG) is indicated in the DRI, therefore, DG was used as RDV for these nutrients.”

Point 6:-Section 2.4 and 2.5: Since the authors have stated the potential maximum score for NRF9.3, I wonder if it is useful to also state the theoretical minimum. For E-DII, could the authors also state the plausible range?

Response 6: Thank you for your insightful comment. It is useful to describe the NRF9.3 of 900 points with full adherence to the dietary intake standards because it is a goal, but also a feasible one. The rules for calculating the NR9 sub-score give an upper limit of 100 points if the qualifying nutrient intake reaches the DRV, so the maximum value can be determined. On the other hand, the calculation rule for the LIM sub-score considers the nutrient intake beyond the DRV and does not provide an upper limit, so the minimum value cannot be determined. In other words, there is no minimum value.

The theoretical full range of DII score using the 45 parameters is -8.87 to 7.98, while the score using the 25-30 parameters falls in the range of -5.5 to 5.5. The last sentence was added in the text (pages 6, lines 254-256)

Point 7:-Line 208: Besides under-reporter, is there any over-reporter?

Response 7: Thank you for your question. There were no over-reporters.

Point 8:-Line 300-309: I am quite confused by this paragraph and the figure, maybe because of the way DII is constructed (higher score is worse), thus making interpretations difficult. I wonder if the authors can think of a better way to present these data. If not, maybe it will help the readers by providing examples (e.g. referring to a specific component score when discussing differences by tertiles).

Response 8: Thank you for your suggestion. We added the explanatory example (page 10, lines 390-393)

“The higher E-DII scores indicate higher inflammatory potential, which is not a desirable dietary state. In Figure 3, parameter-specific E-DII scores are shown. For example, the intake of dietary fiber, one of the anti-inflammatory nutrients, was much lower than the global mean, which resulted in a large positive parameter-specific E-DII score.”

Point 9:-Line 332: Is it possible to distinguish between good and bad oils/fats?

Response 9: Thank you for your question. In order to divide the oils included in the category of fats and oils in the Food Composition Table into good and bad oils, we need standard criteria for which oils and fats are good and which are bad, and as far as we know, there is no consensus on this.

Point 10:-Line 349: This sensitivity analysis must be described in Methods too to prepare the readers.

Response 10: Thank you for your suggestion. Sections 2.7 and 2.8 have been replaced. The title of 2.8 was changed to “Misreported energy intake and sensitivity analysis”.  A sentence below was added in the 2.8 section (page 7, lines 310-311).

“Sensitivity analysis was performed excluding 17 participants with misreported energy intakes.”

This is not something that was pointed out by the reviewers, but please allow us to make the following changes to improve this manuscript. There is a "Intake of vegetables and fruits was positively associated with high NRF9.3 scores and negatively associated with high E-DII scores ~" in the results mentioned in the abstract of this paper, which was derived from both the main analysis and the sensitivity analysis (Tables S6 and S7 in the old manuscript). We have moved the old Tables S6 and S7 to the Results section as Tables 4 and 5 because the content of the abstract should be included in the main text, not in the supplementary material.

Point 11:-Table S4: The categorization of nutrients into energy-producing, vitamins, and minerals is not clear.

Response 11: Thank you for your comments. In order to clarify this point, we added horizontal lines to Table S4.

Point 12:-Line 413: It was mentioned that “The main strength of our study is that it is the first time that the NRF9.3 index been applied to assess the overall dietary quality of pregnant women.” But why is this important? Is NRF9.3 better than other diet quality index, and if so how so?

Response 12: The authors believe that both food-based and nutrient-based dietary indices are useful in assessing the overall quality and properties of a pregnant woman's diet. We do not think that only NRF9.3 is better than other dietary indices. NRF9.3 is a nutrient-based index, and nutrients requirements are similar across different countries. Because the NRF9.3 represents adherence to the national dietary intake standards, the score is easily interpretable and understandable. Therefore, in this study, we used NRF9.3.

Point 13:-Line 430: It was mentioned that seasonal variation was not considered, but I think you have the data as inferred from the Methods, so why not adjust for seasonality in your analyses?

Response 13: According to the reviewer's suggestion, we reanalyzed the results by adding season (four categories) as an adjustment factor. And we found that the results of comparison across the tertiles and all trend tests remained unchanged and unaffected. But we would like to make “the adjustment for seasonality” a subject of the next study using a larger sample size cohort. Therefore, we did not change this part in the manuscript.

Round 2

Reviewer 2 Report

Thank you, the authors have addressed my concerns and edited the manuscript sufficiently, and I have no further comments.